# Preoperative Embolization of Vertebral Metastasis: Comprehensive Review of the Literature

**DOI:** 10.3390/diseases11030109

**Published:** 2023-08-28

**Authors:** Eliodoro Faiella, Domiziana Santucci, Daniele Vertulli, Fabrizio Russo, Gianluca Vadalà, Rocco Papalia, Bruno Beomonte Zobel, Vincenzo Denaro, Rosario Francesco Grasso

**Affiliations:** 1Department of Radiology, University of Rome “Campus Bio-Medico”, Via Alvaro del Portillo, 21, 00128 Rome, Italy; e.faiella@policlinicocampus.it (E.F.); daniele.vertulli@unicampus.it (D.V.); b.zobel@policlinicocampus.it (B.B.Z.); r.grasso@policlinicocampus.it (R.F.G.); 2Department of Orthopaedics, University of Rome “Campus Bio-Medico”, Via Alvaro del Portillo, 21, 00128 Rome, Italy; f.russo@policlinicocampus.it (F.R.); g.vadala@policlinicocampus.it (G.V.); r.papalia@policlinicocampus.it (R.P.); v.denaro@policlinicocampus.it (V.D.)

**Keywords:** angiography, embolization, interventional radiology, secondary vertebral lesions, transcatheter arterial embolization

## Abstract

The aim of this review is to determine the safety and efficacy of pre-operative spinal metastases embolization procedures. Two reviewers independently conducted the literature search (on MEDLINE databases), including in the review of all the studies that used pre-operative TAE to treat spinal metastases. Twelve articles on pre-operative spinal metastases embolization were selected. Most of the studies demonstrated the low complication rate of pre-operative embolization. The most important study strength is that there are very few reviews in the literature with the setting on pre-operative vertebral metastases embolization. A limitation of the review is that the studies included were predominately retrospective case-control studies, increasing the risk of bias in the primary data. Plus, divergent surgical and embolization procedures were performed in the studies, causing a potential risk of bias in the pooled results. We can conclude that preoperative arterial embolization of vertebral metastases is a safe, well-tolerated technique that reduces surgical blood loss and facilitates surgical tumor resection.

## 1. Introduction

Secondary lesions’ spine involvement has a dramatic influence on patients’ quality of life, basically through spine instability, refractory pain, and neurological complications due to spinal cord compression. Patients who can not be treated conservatively should need surgical treatment, which basically consists in improving the patient’s quality of life. There are many different kinds of surgery techniques, but the most typical approach consists of a wide laminectomy and minimally invasive percutaneous transpedicular instrumentation, followed by tumor excision. The orthopedic techniques are improving yearly, which is reflected in the higher life expectance in these patients and, finally, the higher number of spinal metastases. The primitive tumors most commonly and typically spread to vertebrae are breast, lung, kidney and prostate cancer, mainly by the arterial vessels; the metastasis first affects the posterior wall of the vertebral body, with later involvement of the anterior portion, lamina, and pedicles.

Surgical approaches are dramatically complicated by intraoperative blood loss, mostly because vertebral metastases have higher vascularization than normal vertebrae [1]. In recent years, transcatheter arterial embolization (TAE) has been used in the preoperative or palliative management of both primitive and secondary bone tumors [2]. In general, embolization therapy can be defined as introducing a substance into a blood vessel to occlude or reduce the blood flow to a region or organ, in this case, the spinal metastases. In recent years, several studies have reported good results for the TAE approach before surgery for primary hypervascular bone tumors and for secondary bone metastases [3,4]. Regarding techniques, preoperative embolization of vertebral metastases can be performed anywhere along the vertebral column; there are no specific limits. Statistically, the majority of metastases are located in the thoracic and lumbar spine. Regarding the use of embolization of metastases, the aim of this technique is the proximal occlusion of the feeding vessel, not allowing vascular access into the tumor vessel bed, resulting in cell death [5]. The more specific we are to the lesion, the less damage to the surrounding structures we can cause. In order to achieve this, it is important to be very selective with vessel catheterization.

The most frequently used particles in TAE are gelfoam, coils or microcoils, N- butyl-cyanoacrylate, ethylene–vinyl alcohol (Onyx), and Polyvinyl alcohol (PVA). Gelatin sponges (gelfoam) cause temporal occlusion of the tumor vasculature; after 24 h from the embolization, the vessel is recanalized. PVA is available in the form of non-reabsorbable particles; sizes range from 50 to 1200 mm; the 100–300 mm particles are the most commonly used; after PVA embolization, surgery should take place within 7 days of embolization in order to avoid recanalization. Intra-arterial ethanol injection is associated with different kinds of phenomena, including fibrinoid necrosis, intimal sclerosis, angionecrosis, and damage to normal tissue, which definitely cause tumor cell apoptosis. Onyx via either transarterial approach or via direct puncture of the tumor for spinal metastases is an effective and safe procedure; specifically, Onyx 18 and 34 are the most used, which are composed of 6% and 8% ethylene vinyl alcohol copolymer and 94% and 92% dimethylsulfoxide, respectively. In contrast to gelatin sponge and PVA, which cause a temporal vessel occlusion, Onyx is associated with permanent occlusion, so tumor recanalization is not a real problem. After the procedure, tumor softening was observed 8 days after embolization. Considering all these embolic materials, PVA is the most widely used, mainly because of its biocompatibility; however, the irregular PVA particles tend to aggregate, which may lead to catheter occlusion, limiting the degree of penetration into a vascular lesion [6]. The preferred technique is particle embolization, and small-to-medium-sized particles should be used to reduce the amount of blood supply during surgery. Polyvinyl alcohol particles with a diameter of 150 to 250 mm would be a typical option. Embolization is carried out until total stasis is reached [5].

Spinal cord ischemia, which causes a cord infarction and neurologic abnormalities, is the most frequent adverse event associated with the embolization of vertebral body metastases. Because it is so uncommon, the incidence rates of this complication have not been documented. According to the literature, embolization should be postponed if the patient exhibits neurologic symptoms while receiving the intervention, if the spinal arterial supply is found to originate from the same level as the tumor enhancement, or if stable catheter positioning cannot be achieved [7]. Cervical metastases are supplied with blood by the vertebral arteries; in this situation, vertebral embolization is associated with an increased risk of ischemic stroke in the brain’s posterior circulation. Both retrospective and prospective analyses have been used to examine when embolization should occur prior to surgery. Embolization has been shown to be most effective at reducing intraoperative blood loss when performed within 24 h of surgery but no more than 48 h. The efficacy of coils in vertebral metastases remains controversial since they occlude pre-tumoral arterial branches without complete filling of the tumor microvascular structures; moreover, in some tumors such as renal cell carcinoma, the use of coils seems to be ineffective because of their rich vascularity [8].

### Indications

In 1975 Feldman et al. [1] first described TAE as a useful tool in the management of bone tumors, performing two cases of peripheral bone lesions embolization. Since this pioneering study, preoperative embolization has shown good results and high efficacy in the following years. Today it represents the standard of care for hypervascular spinal tumors, especially for renal cell spinal metastases. However, indications for preoperative spinal tumor angiography and endovascular embolization greatly vary among centers, and no guidelines about this procedure and its use are present.

The aim of this comprehensive review of the literature is to explain the modality, advantages, and efficacy of pre-operative spinal embolization procedures.

## 2. Materials and Methods

### 2.1. Guidelines

The present review was performed in accordance with the guidelines of the 2009 PRISMA (preferred reporting items for systematic reviews and meta-analysis) statement.

### 2.2. Data Sources

Two radiologists independently employed MEDLINE databases, such as PubMed and Web of Science, for the research, using the following keywords: “spinal tumours”, “spinal hypervascular tumours”, “spinal metastases”, “preoperative endovascular embolization”, “preoperative transarterial embolization”, “preoperative embolization”, “endovascular embolization” and “transarterial embolization”. In addition, the references of the eligible studies were screened manually to identify additional eligible studies.

### 2.3. Inclusion and Exclusion Criteria

The inclusion criteria were as follows:(1)Studies where TAE was used to treat spinal metastases pre-operatively;(2)Studies evaluated outcomes: intraoperative blood loss, transfusion requirement, operative time, overall survival, and complication rate.

The publication language was restricted to English.

The exclusion criteria were as follows:(1)Duplicated studies were eliminated;(2)Studies where TAE was performed to treat primitive bone lesions;(3)Studies where TAE was performed after surgical intervention.

The study search and selection are shown in Figure 1.

The characteristics of the included studies are summarized in Table 1.

## 3. Results

The level of evidence of this review is level V, evidence from systemic reviews of descriptive and qualitative studies.

### 3.1. Time to Surgery

Barton et al. [5] confirmed that surgery ideally should be performed immediately after embolization to prevent revascularization, and in any case, it should be performed no later than 3 days thereafter.

Tan et al. [3] reported that the median EBL (Estimated Blood Loss) was the lowest (1000 mL) in the patients who underwent surgery between 13 and 24 h, although the differences were not statistically significant. Indeed the recommendation is to perform surgery as early as possible from TAE to benefit from the effect of tumors’ blood supply devascularization.

### 3.2. Complications

Tan et al. [3] reported that the median EBL was 1000 mL in groups who received ‘gelfoam only’ and ‘gelfoam + PVA’ and 1500 mL in the group receiving ‘coil + particles’, even if no statistical significance was found between these groups. In embolized patients with primary spine tumors, thyroid metastases, and spinal metastases from HCC, there was a trend for reduced median blood loss; however, the difference was not statistically significant. Preoperative hemoglobin, kind of surgery, tumor type, and operating duration were all found to be significant predictors of either blood loss or the need for transfusions in univariate analysis. After correcting for the relevant factors, multivariate linear regression revealed no significant differences between the embolized and non-embolized groups in terms of blood loss or transfusion units.

Barton et al. [17] reported an intraoperative EBL of 500–1500 mL in patients who had undergone preoperative embolization and of 2000–18,500 mL in patients who did not receive TAE.

Considering 16 patients with metastasis from renal cell carcinoma (only 2 of them were vertebral), Sun et al. [9] obtained an average intraoperative blood loss of 533 mL; a significant EBL reduction was observed when more than 70% of tumor vascularity was embolized (460 vs. 750 mL).

Wirbel et al. [11] reported an average intraoperative post-TAE EBL of 1650 mL for 21 spinal lesions. The total survival time did not differ between the two groups: 12.2 months in the embolization group and 12.0 months in the group that was not embolized (ranges: 4–34 months and 3-36 months, respectively).

Rehak et al. [12], after drastic surgery for RCC spinal metastases, conducted a retrospective investigation on 15 patients. EBL was higher in the 8 patients who had been embolized (4750 mL) than it was in the non-embolized group of 7 individuals (1786 mL). Despite these results, there were significant differences in the size of the tumor and the intricacy of the approach between the embolized and non-embolized groups, preventing a fair comparison from being made. They finally concluded that metastasis size, tumor extension, the technical complexity of the surgery, and the efficacy of preoperative embolization affect the amount of EBL.

In 2008, in work by Kickuth et al. [10], the efficiency of transcatheter arterial embolization (TAE) of hypervascular metastatic lesions of the bone prior to orthopedic excision and stabilization was to be evaluated in connection to intraoperative estimated blood loss (EBL). They classified TAE devascularization of 22 vertebral lesions into three grades: grade 1 (>75% reduction of tumor blush), grade 2 (50–75% reduction of tumor blush), and grade 3 (<50% reduction of tumor blush). TAE efficacy was related to the devascularization grade: in patients with grade 1 the median EBL was 500 mL; in grade 2 the median EBL was 1475 mL and in grade 3 the median EBL was 2500 mL. No correlation was found between EBL and operating time or between average maximal tumor size and EBL. Minor and major complications during and after TAE were 4.5% and 9%, respectively.

In 2012 Kato et al. [13] reported a statistically significant difference in EBL between the embolization group (520 mL) and the non-embolization group (1128 mL). In the embolization group, no correlation was found between the EBL and the degree of tumor vascularization, completeness of embolization, or time between embolization and surgery.

Zaborovskii et al. [16] retrospectively compared, in a case-control study of 54 patients, the efficiency of different bleeding control methods and their influence on surgical outcomes and survival rates after RCC spinal metastases decompression. Moreover, 32 individuals who received preoperative tumor embolization were part of the first group (EMB). Additionally, 22 patients in the second group (HEM) received surgical care while being administered intraoperative local hemostatic medications. The EMB group’s median EBL (1275 mL) was lower than the HEM group’s (1400 mL) median without significant differences. The postoperative drainage loss in the HEM group (250 mL) was significantly less than that in the EMB group (500 mL). The complication rates (infections, hematomas, neurological deficit) were nearly equal in all groups. They came to the conclusion that not all patients with MRCC require preoperative embolization because there are other options for controlling bleeding, such as using contemporary hemostatic medications.

Clausen et al. [18] in 2015 showed that between the control group (735 mL) and the embolization group (618 mL), there were no appreciable differences in EBL. Blood loss in the embolization group was reduced significantly (*p* = 0.041) from 902 mL (SD, 416 mL) to 645 mL (SD, 289 mL) in the subanalysis of hypervascular metastases. Additionally, the embolization group’s procedure took considerably less time than the control group’s. From a median of 124 min (range: 80–183 min) to 90 min (range: 54–252 min), there was a 27% drop. Atrial fibrillation and pleural exudates were two of three mild complications that were recorded after surgery. A right common femoral artery thrombosis developed during angiography was a major complication.

Hong et al. [15] studied 52 patients who underwent palliative decompression for spinal metastasis of renal carcinomas. Of the 52 patients, 23 (or 44%) had tumors that were hypervascular. The patients with hypervascular tumors showed more intraoperative blood loss. EBL was greater in the non-embolization group (1988 mL) than in the embolization group (1095 mL) in cases of hypervascular tumors. No statistically significant data about clinical outcomes were found based on the primitive tumor and on the site of the metastasis. The most common were pulmonary (7/52, 13.5%) and wound complications (6/52, 11.5%). Four patients (7.7%) had wound dehiscence, repeated debridement, and advancement flap operations.

Tan et al. [3] studied a total of 221 patients, of which 48 (22%) were embolized and 173 (78%) were not. More than half of the embolized patients (61%) were able to achieve total embolization (>80% reduction in tumor blush). Patients with entire embolization had median EBLs of 900 mL, which was considerably (*p* = 0.05) lower than patients with partial (50%) and subtotal (50–80%) embolization (1600 mL and 1350 mL, respectively) embolization.

The median EBL was statistically lower in the proximally embolized group than the distally embolized group (800 vs. 1200 mL) since the proximal embolization could occlude small caliber collateral vessels supplying the tumor.

They found a statistically significant increase in median EBL in embolized cases of breast metastases.

The type of surgery has no effect on the median EBL. It is interesting to note that in embolized cases of thoracolumbar posterior instrumentation and decompression, EBL increased statistically significantly.

Their entire embolization rate was obtained in two-thirds of their cases, and these cases show significantly decreased EBL. As a result, they advise that the goal in every case having pre-operative embolization be total embolization.

### 3.3. Effectiveness

Robial et al. [14] compared the EBL in embolized versus non-embolized patients depending on the primary tumor and the extent of surgery. The primitive cancers were: 28 breast cancer (30.1%), 19 pulmonary carcinoma (20.4%), 16 renal cell carcinoma (17.2%), and 30 other cancers (32.3%). Surgical procedures were: 52 thoraco-lumbar laminectomies with instrumentation, 29 thoraco-lumbar corpectomies or vertebrectomies, 12 cervical corpectomies. No statistically significant difference was found among the different primitive cancer in EBL. All of the histology samples showed statistically significant increases in bleeding depending on the type of surgery (corpectomy/vertebrectomy versus thoracolumbar instrumentation and cervical corpectomy): breast (1775 mL vs. 778 mL), pulmonary (2500 mL vs. 430 mL), renal (3346 mL vs. 1175 mL), and others (1550 mL versus 474 mL).

When surgical resection is intended, the current standard of care is to do a diagnostic spinal angiography, followed, when possible, by embolization, in all patients with RCC spine metastases. Compared to non-embolized tumors, even partial embolization of RCC metastatic lesions appears to lessen blood loss.

## 4. Discussion

Nowadays, TAE is largely used in the preoperative or palliative management of secondary bone metastases [2]. In fact, most studies report that EBL and complications are lower after pre-surgical TAE compared to surgery performed without TAE. The factors associated with the effectiveness of pre-operative embolization remain unclear.

Even more, the overall rate of complete embolization reported by studies varies between 43.8% and 90.5% [11,13,18], with the main causes of incomplete embolization represented by the presence of radiculomedullary artery, catheter-induced dissection of the feeding artery, the presence of hypoplastic arteries and a blood supply from multiple vertebral arteries.

In a recent meta-analysis, however, Zhong-yu et al. [19] found that the embolization effectiveness on intraoperative blood loss and transfusion requirements amount was statistically significant only for hypervascular tumors. In another older review and meta-analysis, Luksanapruksa et al. [8] reported that the embolization group was associated with slightly less intraoperative blood loss, even if without a statistically significant difference. Moreover, the overall survival and complication rates were equal in the embolization and non-embolization groups. The same review reported that in the embolization group, the blood loss found by drainage was greater than that in the non-embolization group, but the study explained that the reduced blood loss could be due to the additional application of intraoperative local hemostatic agents. Generally, the use of antifibrinolytic agents and controlled deliberate hypotension has been shown to reduce perioperative blood loss and transfusion requirements in patients undergoing spine surgery. However, the use of these local hemostatic agents might have been related to the development of wound hematomas. Precisely, because of these sometimes very different results, intraoperative blood loss cannot always be considered a parameter of clinical efficacy.

EBL is mostly related to the residual tumor after embolization than to the tumor size [5]; hence as many tumor feeders as possible should be found and embolized.

Very few studies reported complication rates, basically due to the small sample size; Kickuth et al. [10] reported a complication rate of 4.5-9%; considering all the literature, the complications included both symptomatic [7] and asymptomatic [20] cerebellar infarction; permanent paralysis and paresthesias [21]; spinal infarction resulting in permanent sensory deficit after a microcoil which ended up in the right sulco-commisural artery [22]; cord ischemia following embolization resulting in persistent bilateral lower limb weakness [23]; epidural hemorrhage [24]. As reported in the work of Hong et al. [15], contiguous ligation of three segmental vessels bilaterally did not affect the neurologic system, according to an animal investigation. However, a dog model with >4 layers of ligation showed ischemic cord impairment. We thus thought it safe to execute embolization up to two levels bilaterally and up to three levels unilaterally.

Then, we may conclude that embolization appears to be a safe procedure with low complication rates.

Regarding embolizing particles, we found that microspheres might be preferable to achieve complete embolization and to further delay tumor revascularization [4]. A measure of precaution is to use mid-size particles ranging from 300 to 500 mm; smaller particles may inadvertently go through a non-visualized micro-anastomosis with prominent vessels. Larger particles will quickly clog proximal vessels or will clump into the microcatheter.

The use of materials should also be related to the specific patient to treat. Pikis et al. [2] observed that older patients’ arteries were more convoluted and challenging to catheterize, which limited the possibility of a full embolization. As reported in their local experience, due to technical restrictions (vascular tortuosity) or predicted difficulties (near origin of anterior spinal artery or short cervicothoracic feeders), embolization was not carried out in 6% of patients. Unsurprisingly, they found that the more convoluted and challenging-to-catheterize arteries of elderly individuals made full embolization more difficult. Additionally, bigger tumors fed by many arterial sources, metastatic lesions in the cervical and upper thoracic spine, and other dangers and limitations unique to the embolization procedure all reduced the effectiveness of embolization.

Ladner et al. [25] employed Onyx as an embolizing agent, demonstrating how this fluid agent can improve the penetration of the tumor parenchyma, in particular by associating the use of the double balloon catheter. These authors demonstrated how this technique leads to a significant reduction in post-operative bleeding in an innovative way reducing the EBL from 2400 ± 738 mL to 584 ± 124 mL of the control group.

The riskiest embolization procedures are those of metastatic lesions located in the cervical and upper thoracic spine and lesions fed by multiple arterial sources, resulting in a limited efficacy of the technique. Longitudinal and peri-vertebral anastomosis is the most dangerous because the migration of particles may take place via these anastomoses to an intercostal/lumbar artery at the upper or lower level even when the embolization is strictly carried out through a radicular artery that does not give birth to a radiculomedullary artery, or to a contralateral artery, feeding a radiculo-pial or radiculo-medullary artery. Because of that, selective micro-angiographic injections must always precede embolization in order to reduce possible complications in identifying target vessels and the most potentially dangerous anastomoses.

New agents, such as hydrophilic, non-resorbable, collagen-coated, acrylic microspheres, continue to be developed to resolve these difficulties.

The majority of studies focused on RCC metastasis, whose embolization is particularly useful and effective, basically because of their elevated risk of hemorrhage [7,26].

Our review systematically analyzes the papers present in the literature on preoperative vertebral embolization, exploring in detail not only the postoperative blood loss and the complication rate but also the correlation of these with the main technical aspects of the methodical.

This review has some important limitations. First, the studies included were predominately retrospective case-control studies, increasing the risk of bias in the primary data. Second, we excluded studies without available and studies written in languages other than English. Third, divergent surgical and embolization procedures were performed in the studies, causing a potential risk of bias in the pooled results. Fourth, different kinds of metastasis were included in the studies, causing a risk of bias in the results.

Based on our personal experience, a careful evaluation of the CT or MRI preoperative imaging, preferably angiography CT, is essential for identifying the main vessels afferent to the lesion. The target of the embolization must be carefully evaluated based on the surgery aim, the general conditions of the patient, and the complexity of the anatomical site. It is necessary to carry out the preliminary angiographic evaluation with a pump run for correct mapping of the tumor vascular supply and to identify, in critical sites, the origin of Adamkiewicz’s artery or dangerous anastomosis. Also, the choice of the embolizing agent must be carefully assessed considering the objective of the embolization, the control of postoperative hemostasis, and the risk of embolization in non-target locations. We reported our own case in Figure 2.

In conclusion, we can resume that preoperative arterial embolization of vertebral lesions is a safe, well-tolerated technique that reduces surgical blood loss and facilitates surgical tumour resection.

Further studies on larger samples are needed to evaluate the correct indication of the different embolizing agents. Longitudinal studies on specific populations are required to evaluate the effectiveness of these techniques on different lesions and to obtain standardized indications.

## Figures and Tables

**Figure 1 diseases-11-00109-f001:**
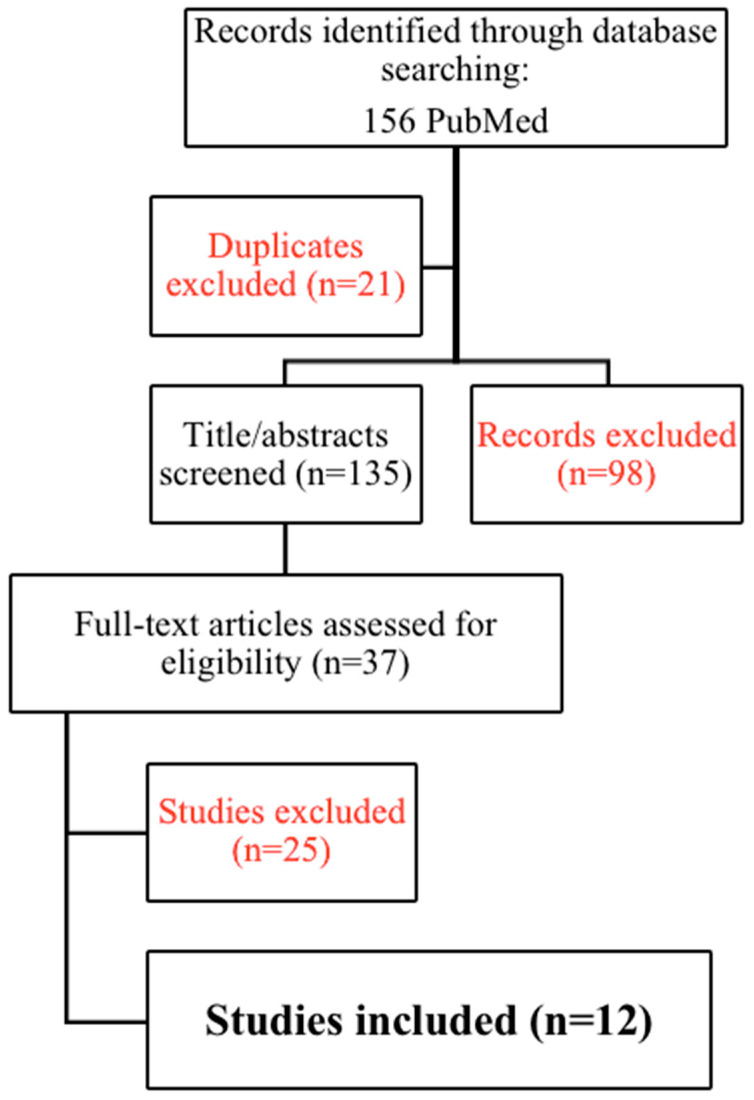
PRISMA (preferred reporting items for systematic reviews and meta-analysis) 2009 flowchart for study search and selection.

**Figure 2 diseases-11-00109-f002:**
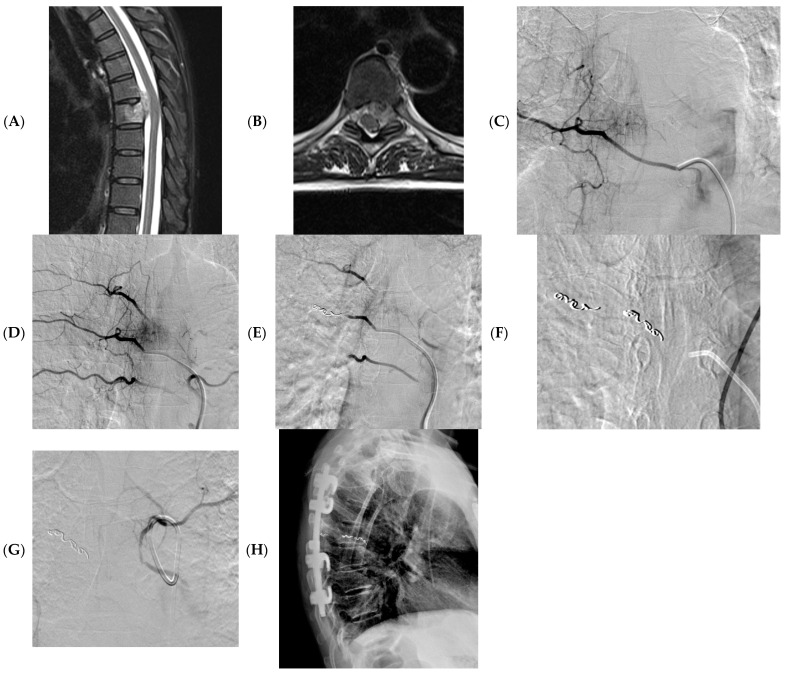
A 58 years-old woman with a sarcoma vertebral metastasis. MRI (axial and sagittal T2 images) showing pathological tissue at T6-7 with involvement of the vertebral canal and partial medullary cord compression (**A**,**B**). Pre-operative angiography showing tissue vascularization, including Adamkiewicz artery (**C**,**D**); embolization was obtained using microparticles (300–500 microns) and microcoils (**E**–**G**); X-ray post-surgery (**H**).

**Table 1 diseases-11-00109-t001:** Studies reporting vertebral metastases pre-operative embolization ad their characteristics. Notes: GCT: giant cell tumor; ADK: adenocarcinoma; HCC: hepatocellular carcinoma; RCC: renal cell carcinoma; LC: lung cancer; EBL: estimated blood loss.

Author	Year	Number of Patients with Spinal Metastasis	Tumor Type	Location	Embolic Material	Interval from Embolization to Surgery	Complications	EBL
**Feldman** [1]	1975	0	GCT, ADK		Gelfoam (methylcellulose) strips cut into 2-3 mm pieces, soaked in Renografin 76	1–2 weeks	No data	No data
**Barton** [5]	1996	11	RCC	Thoracic, 3	Synthetic tissue adhesive (Histoacryl mixed with Lipiodol at a ratio of between 1:3 and 1:5) or a combination of equine collagen flocculi and stainless-steel coils or the equine collagen flocculi used alone in nine cases or polyvinyl alcohol foam particles (Ivalon; Drivalon 300-600 pm)	1–2 days	No data	EMB group: 500-1500 ml
			TC	Lumbar, 8				
			Breast					Non EMB group: 2000–18,500 mL
			Utherus					
			ADK					
**Sun** [9]	1998	2	RCC	T12, 1	510-1,000 μm PVA particles or mini Gianturco stainless-steel and Hilal titanium microcoils (Cook)	24–120 h	Fever 6 hours after the procedures (3 patients)	533 mL
				L5-S1, 1				
**Wirbel** [10]	2005	21	RCC	No data	Thrombogenic platinum coils or Contour Emboli 250-to 350-μm particles	<24 h	No data	EMB group: 1650 mL
			Thyroid					
			Breast					Non EMB group: 3880 mL
**Kickuth** [11]	2008	1	RCC	L2	PVA particles (150–300 μm and 350–500 μm)	<24 h	4.5% minor complications and 4.5 % major complications (gluteal abscess)	500–2500 mL
**Rehak** [12]	2008	15	RCC	Cervical, 3	Microparticles 350–500 μm and 500–700 μm in size		One embolized patient died as a consequence of haemor- rhagic shock, blood loss in total of 7000 ml occurred during this operation and adequate volume resuscitation was not achieved	1786 mL in Non embolized group and 4750 mL in Embolized group
				Thoracic, 8				
				Lumbar, 3				
				Sacral, 1				
**Kato** [13]	2012	46			Polyvinyl alcohol particles, gelatin sponge, and metallic coils	<3 days		EMB group: 520 mL
								Non EMB group: 1128 mL
**Robial** [14]	2012	93	Breast, 28	Cervical, 27 lesions	Microspheres with a diameter ranging from 500 to 700 microm (Embosphere®, BioSphere Medical, Rockland, MA)	<48 h	No data	Prendi dal testo
			Lung, 19	Thoracic, 76 lesions				
			Kidney, 16	Lumbar, 29 lesions				
			Others, 30					
**Pikis** [2]	2014	96	RCC, 22					
			Breast, 6					
			Thyroid, 3					
			Other, 3					
**Hong** [15]	2017	52	HCC 12	T6 9	polyvinyl alcohol (PVA) particles and/or gelatin sponge (Gelfoam)	<48 h	Pulmonary problems (7/52), wound problems (seroma, 6/52), wound dehiscence (4/52)	1988 mL in Non embolized group and 1095 mL in Embolized group
			RCC 10	T3 8				
			LC 9	T4 7				
**Tan** [3]	2017	209	Renal, 14		Gel foam slurry or polyvinyl alcohol (PVA)	<48 h		
			HCC, 7					
			Thyroid, 5					
			Lung, 65					
			Breast, 42					
			Gastrointrstinal, 19					
			Others, 57					
**Zaborovskii** [16]	2018	54	RCC	Thoracic, 33	Gelatin sponge particles	<48 h	Infections and hematoma in the site of wound	1275 mL for EMB group, 1400 mL for HEM group (intraoperative local hemostatic agents)
				Lumbar, 21				

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
