# Peer review of "Preoperative Embolization of Vertebral Metastasis: Comprehensive Review of the Literature"

_diseases, 2023, doi:10.3390/diseases11030109_

Round 1

Reviewer 1 Report

This is an interesting review of the literature reporting pre-operative transcatheter arterial embolization of vertebral metastases and their characteristics. Only 12 studies were included and most of them reported that bleeding complications were lower after pre-surgical transcatheter arterial embolization as compared with surgery performed without embolization.

As stated by the authors, studies analyzed in this review were mostly retrospective and techniques of surgery and embolization were heterogeneous.

Minor point:

Surprisingly, the first study cited in table 1 included 0 patient with spinal metastasis.

Author Response

Reviewer 1

This is an interesting review of the literature reporting pre-operative transcatheter arterial embolization of vertebral metastases and their characteristics. Only 12 studies were included and most of them reported that bleeding complications were lower after pre-surgical transcatheter arterial embolization as compared with surgery performed without embolization.

As stated by the authors, studies analyzed in this review were mostly retrospective and techniques of surgery and embolization were heterogeneous.

Minor point:

Surprisingly, the first study cited in table 1 included 0 patient with spinal metastasis.

Thanks to the Reviewer for this observation. Actually, the first study cited was the first paper about vertebral lesions, so we decided to include it for historical reasons discussing the embolization materials used.

Reviewer 2 Report

I would like to take the present time to congratulate the authors for conducting the present research. However I would like also to share a few comments and concerns:

The Abstract should be improved since it provides basically no relevante information regarding the manuscript major take always. The Abstract does not follow the PRISMA by no means as the authors state in the text. 

The Abstract subheadings are not required

I recommend the authors to place the keywords by alphabetic order

I suggest the authors to increase the keywords to 5 in order to increase the manuscript reach

There is no aim sentence in the end of the Introduction and the study rational should be improved. Why do we need this review?

The exclusion criteria should be re-written. Exclusion criteria is not the opposite of inclusion criteria. 

I cannot find the registration number as requested by PRISMA

Did the authors contact authors from previous studies?

Did the authors conducted a Manuel search?

The summary table requires que proper reference for each study (Table 1)

Did the authors conducted any Risk of Bias assessment?

Study strength is missing

Levels of evidence is missing

The reference list is not according to authors instruction. 

Author Response

I would like to take the present time to congratulate the authors for conducting the present research. However I would like also to share a few comments and concerns:

The Abstract should be improved since it provides basically no relevante information regarding the manuscript major take always. The Abstract does not follow the PRISMA by no means as the authors state in the text.

Thanks to the Reviewer for this important suggestion. We modified the abstract following PRISMA instruction.

The Abstract subheadings are not required

The subheadings have been removed as suggested

I recommend the authors to place the keywords by alphabetic order

Thanks to the Reviewer for the indication. The keywords were ordered

I suggest the authors to increase the keywords to 5 in order to increase the manuscript reach

The keywords were added as suggested (“angiography; embolization; interventional radiology; secondary vertebral lesions; transcatheter arterial embolization”)

There is no aim sentence in the end of the Introduction and the study rational should be improved. Why do we need this review?

Thanks to the Reviewer for the suggestion. The aim was reported “The aim of this comprehensive review of the literature is to explain the modality, the advantages and efficacy of pre-operative spinal embolization procedures.”

The exclusion criteria should be re-written. Exclusion criteria is not the opposite of inclusion criteria.

The exclusion criteria were written as follows: “The inclusion criteria were as follows:

1)        studies where TAE was used to treat spinal metastases pre-operatively;

2)        studies which evaluated as outcomes: intraoperative blood loss, transfusion requirement, operative time, overall survival, and complication rate.

The publication language was restricted to English.

The exclusion criteria were as follows:

1)        duplicated studies were eliminated;

2)        studies where TAE was performed to treat primitive bone lesions;

3)        studies where TAE was performed after surgical intervention.

I cannot find the registration number as requested by PRISMA

Thanks to the Reviewer for this comment, unfortunately, we tried to register the study on PROSPERO, but PROSPERO does not accept scoping reviews, literature reviews or mapping reviews. The same website invites the authors to not stop from submitting full protocol or completed review for publication in a journal.

Did the authors contact authors from previous studies?

No, we did not.

Did the authors conducted a Manuel search?

Yes, we did.

The summary table requires que proper reference for each study (Table 1)

Thanks. The reference has been added for each study.

Did the authors conducted any Risk of Bias assessment?

No, we did not.

Study strength is missing

Thanks for this comment. The strengths points of the study have been added in the Discussion paragraph “Our review systematically analyzes the papers present in literature, on preoperative vertebral embolization, exploring in detail not only the postoperative blood loss and the complication rate, but also the correlation of these with the main technical aspects of the methodical.”

Levels of evidence is missing

The level of evidence of this review is level V, evidence from systemic reviews of descriptive and qualitative studies. A sentence was added in the results section as suggested.

The reference list is not according to authors instruction.

Thanks to the Reviewer for this comment. We followed authors instruction at this website link: https://www.mdpi.com/authors/references, and we check again all the references.

Reviewer 3 Report

In this manuscript, Faiella and colleagues conducted a comprehensive review of literature to evaluate safety and efficacy of preoperative embolization of vertebral metastasis. The authors selected a total of 12 published studies related to the subject for analysis and concluded that preoperative embolization is safe and well-tolerated for reduction of blood loss during surgical tumor resection. The whole manuscript is well-written and easy to understand, with methods and results presented in detailed and clear manners.

There are some minor issues that should be addressed to improve the manuscript.

1.      The introduction is well-written and full of helpful information to understand the embolization procedure. However, the last five paragraphs are much shorter in length compared with previous paragraphs, which is out of balance and symmetry. It is recommended to reorganize these paragraphs.

2.      For Table 1, the table can be organized to look better and more readable.

3.      The place where the indications in lines 131-138 are presented seems to be confusing. It should either be placed in the introduction or the results section.

4.      For Figure 2, there is no description in the results or discussion section. The authors should make necessary descriptions about Fig 2 in the manuscript.

5.      In the lines 225-240 about effectiveness, the authors mentioned that “no statistically significant difference was found about EBL between embolized and non-embolized groups in all the primitive cancer.” However, the authors concluded that preoperative embolization reduced surgical blood loss in the discussion. How did the authors reach the conclusion from the literature review? Also, the description in lines 232-236 is difficult to understand.

6.      In line 290, “reducing the EBL from 584±124 mL to 2400±738 mL of the control group” should be “reducing the EBL from 2400±738 mL of the control group to 584±124 mL”.

7.      There are many grammatic errors and spelling mistakes present in the manuscript. The authors should correct these errors and mistakes in the revised manuscript.

Overall, the English language is in good quality. However, there are many grammatic errors and spelling mistakes present in the manuscript. The authors should correct these errors and mistakes in the revised manuscript.

Author Response

In this manuscript, Faiella and colleagues conducted a comprehensive review of literature to evaluate safety and efficacy of preoperative embolization of vertebral metastasis. The authors selected a total of 12 published studies related to the subject for analysis and concluded that preoperative embolization is safe and well-tolerated for reduction of blood loss during surgical tumor resection. The whole manuscript is well-written and easy to understand, with methods and results presented in detailed and clear manners.

There are some minor issues that should be addressed to improve the manuscript.

  1. The introduction is well-written and full of helpful information to understand the embolization procedure. However, the last five paragraphs are much shorter in length compared with previous paragraphs, which is out of balance and symmetry. It is recommended to reorganize these paragraphs.

Thanks to the Reviewer for the comment. We changed the organization of the paragraphs in the introduction as suggested.

  1. For Table 1, the table can be organized to look better and more readable.

Thanks for this suggestion. The table was charged as figure to make it more readable and the excel was attached as separate file

  1. The place where the indications in lines 131-138 are presented seems to be confusing. It should either be placed in the introduction or the results section.

Thanks for this suggestion. The subparagraph was moved in the introduction.

  1. For Figure 2, there is no description in the results or discussion section. The authors should make necessary descriptions about Fig 2 in the manuscript.

Thanks for this comment. We improved the discussion section with our personal experience and reported the figure 2 as example.

  1. In the lines 225-240 about effectiveness, the authors mentioned that “no statistically significant difference was found about EBL between embolized and non-embolized groups in all the primitive cancer.” However, the authors concluded that preoperative embolization reduced surgical blood loss in the discussion. How did the authors reach the conclusion from the literature review? Also, the description in lines 232-236 is difficult to understand.

Thanks for the indication. The sentence had a different meaning. It has been changed as follows: “No statistically significant difference was found among the different primitive cancer in EBL”

In line 290, “reducing the EBL from 584±124 mL to 2400±738 mL of the control group” should be “reducing the EBL from 2400±738 mL of the control group to 584±124 mL”.

Thanks for the comment, we did the correction.

There are many grammatic errors and spelling mistakes present in the manuscript. The authors should correct these errors and mistakes in the revised manuscript.

Thanks for the comment; we did some corrections as suggested.

Reviewer 4 Report

Dear Authors

my compliments for your paper.

I have only a suggestion. Did you perform this procedure? Did you have some personal opinion about that procedure, some techincal suggestion?

Probably you can improve discussion with these considerations

Author Response

Dear Authors

my compliments for your paper.

I have only a suggestion. Did you perform this procedure? Did you have some personal opinion about that procedure, some techincal suggestion?

Probably you can improve discussion with these considerations

Thank you for this comment that allows us to deepen this important aspect. In our hospital we perform this type of procedure and we can surely propose some suggestions. We report the following sentence in the discussion paragraph: "basing on our personal experience, a careful evaluation of the CT or MRI preoperative imaging, preferably angiography CT, is essential for identify the main vessels afferent to the lesion. The target of the embolization must be carefully evaluated based on the surgery aim, the general conditions of the patient and the complexity of the anatomical site. It is necessary to carry out the preliminary angiographic evaluation with pump-run for a correct mapping of the tumor vascular supply and to identify, in critical sites, the origin of Adamkiewicz' artery or dangerous anastomosis. Also the choice of the embolizing agent must be carefully assessed considering the objective of the embolization the control of post-operative hemostasis and the risk of embolization in non-target locations.”

Round 2

Reviewer 1 Report

The authors have adequately responded to the issues raised and have revised the manuscript accordingly.

Reviewer 2 Report

Dear authors, I gave no further comments.